# Mid-Infrared Reflectance Spectroscopy for Estimation of Soil Properties of Alfisols from Eastern India

Kuntal M. Hati [1], Nishant K. Sinha [1,*], Monoranjan Mohanty [1], Pramod Jha [1], Sunil Londhe [2], Andrew Sila [3], Erick Towett [3], Ranjeet S. Chaudhary [1], Somasundaram Jayaraman [1], Mounisamy Vassanda Coumar [1], Jyoti K. Thakur [1], Pradip Dey [1], Keith Shepherd [3], Pankaj Muchhala [1], Elvis Weullow [3], Muneshwar Singh [1], Shiv K. Dhyani [2], Chandrashekhar Biradar [2], Javed Rizvi [2], Ashok K. Patra [1] and Suresh K. Chaudhari [4]

1 ICAR-Indian Institute of Soil Science, Nabibagh, Berasia Road, Bhopal 462038, India; kuntal.hati@icar.gov.in (K.M.H.); manoranjan.mohanty@icar.gov.in (M.M.); pramod.jha@icar.gov.in (P.J.); ranjeet.chaudhary@icar.gov.in (R.S.C.); somasundaram.j@icar.gov.in (S.J.); mv.coumar@icar.gov.in (M.V.C.); jyoti.thakur@icar.gov.in (J.K.T.); pradip.dey@icar.gov.in (P.D.); pankajmuchhala05@gmail.com (P.M.); muneshwar.singh@icar.gov.in (M.S.); ashok.patra@icar.gov.in (A.K.P.)
2 CIFOR-ICRAF, New Delhi 110001, India; s.londhe@cgiar.org (S.L.); s.dhyani@cgiar.org (S.K.D.); c.biradar@cgiar.org (C.B.); j.rizvi@cgiar.org (J.R.)
3 World Agroforestry (ICRAF), Nairobi 999070, Kenya; a.sila@cgiar.org (A.S.); e.towett@cgiar.org (E.T.); k.shepherd@cgiar.org (K.S.); e.weullow@cgiar.org (E.W.)
4 Indian Council of Agricultural Research, New Delhi 110001, India; ddg.nrm@icar.gov.in
* Correspondence: nishant.sinha76211@gmail.com or nishant.sinha@icar.gov.in

**Abstract:** Mid-infrared (MIR) spectroscopy is emerging as one of the most promising technologies, as it is a rapid and cost-effective alternative to routine laboratory analysis for many soil properties. This study was conducted to evaluate the potential of mid-infrared spectroscopy for the rapid and nondestructive measurement of some important soil properties of Alfisols. A total of 336 georeferenced soil samples from the 0–15 cm soil layer of Alfisols that were collected from the eastern Indian states of Odisha and Jharkhand were used. The partial least-squares regression (PLSR), random forest, and support vector machine regression techniques were compared for the calibration of the spectral data with the wet chemistry soil data. The PLSR-based predictive models performed better than the other two regression techniques for all the soil properties, except for the electrical conductivity (EC). Good predictions with independent validation datasets were obtained for the clay and sand percentages and for the soil organic carbon (SOC) content, while satisfactory predictions were achieved for the silt percentage and the pH value. However, the performance of the predictive models was poor in the case of the EC and the extractable nutrients, such as the available phosphorus and potassium contents of the soil. Specific regions of the MIR spectra that contributed to the prediction of the soil SOC, the pH, and the clay and sand percentages were identified. The study demonstrates the potential of the MIR spectroscopic technique in the simultaneous estimation of the SOC content, the sand, clay, and silt percentages, and the pH of Alfisols from eastern India.

**Keywords:** mid-infrared spectroscopy; soil properties; alfisols; partial least squares; support vector machine; random forest

## 1. Introduction

Soil, which is a complex mixture of minerals, organic matter, microorganisms, air, and water, is one of the most important factors/resources that control agricultural productivity and ecosystem functioning. The preservation and sustainable management of soil resources are essential in tackling humanity's major challenges that are related to food security, climate change, environmental degradation, water scarcity, and biodiversity [1]. Monitoring the factors that control the preservation and sustenance of soil resources is essential for improving and sustaining agricultural productivity and for maintaining the soil health.

However, one of the main challenges in monitoring this resource is the heterogeneity, or spatial variability, of the soil properties. This entails the analysis of a large number of soil samples in order to obtain a comprehensive idea about the soil health and its spatial distribution, which is imperative for formulating strategies to ensure efficient management. Traditionally, soil health information is obtained through conventional laboratory analysis, which is time-consuming, labor-intensive, and, sometimes, not friendly to the environment owing to the use of some toxic chemicals during laboratory-based wet chemistry analysis [2]. Conventional laboratory analysis generally requires separate tests for the different soil properties, as well as a wide range of equipment, which renders them economically nonviable in many cases. To counter these problems, alternate methods of soil analyses that use a single preparation and a spectral scan to predict many soil properties have been developed [3]. Diffuse reflectance infraredFourier transform (DRIFT) spectroscopy is an evolving technology that is based on the interaction of electromagnetic energy with matter, and it provides a great opportunity for the speedy, cost-effective, and nondestructive characterization of the soil composition. This technique has opened up new possibilities for its application in site-specific nutrient management, the monitoring of the soil health in landscapes [4], and for digital soil mapping [5]. Among the different spectroscopic techniques in vogue, DRIFT spectroscopy using the mid-infrared (MIR) region (4000–400 cm$^{-1}$, equivalent to a wavelength of 2500–25,000 nm), provides a good alternative with which to enhance and support the conventional methods of soil analysis, as it reduces some of their limitations, particularly where a high spatial density is needed [6].

Considerable progress has been made in the last two decades in the use of MIR spectroscopy, with the development of new instrumentation and modeling techniques, which have lessened some of its earlier limitations. There has also been improvement in the computational power and the development of more robust statistical tools that can more precisely relate the variability of the spectrum to the variability of the soil characteristics [7]. One of the advantages of infrared spectroscopy (IR) is that it can simultaneously characterize various chemical and physical soil constituents from a single spectrum. Among the different infrared spectroscopic techniques in use, the instruments that use the MIR region are generally considered more useful than those that use the near-infrared (NIR) region of the electromagnetic spectrum, as the MIR region is dominated by fundamental vibrations, whereas the NIR region is dominated by much weaker and broader signals from vibration overtones and combination bands [8,9]. For instance, Pirie et al. [10] observed the greater usefulness of MIR spectroscopy compared to UV–VIS–NIR spectroscopy to predict the soil pH, the contents of organic C, the clay, the sand, and the cation exchange capacity (CEC) for some Alfisols in southeastern Australia. Shepherd et al. [11] report that IR spectroscopy was more repeatable than standard laboratory wet chemistry methods, as the IR-based method almost halved the measurement standard deviation (SD).

Clay minerals and organic carbon are the principal constituents of soils, and they have well-defined fundamental absorption regions in the MIR range. The MIR multivariate calibrations are, consequently, more robust than the NIR for the characterization of the soil properties [7]. Molecular vibrations that are related to alkyl groups, protein amides, carboxylic acids, the associated water, carboxylate anions, and the aromatic groups that are present in the soil organic matter provide opportunities for estimations ofthe-soil organic carbon (SOC) and the total nitrogen (N), with considerable accuracy by MIR spectroscopy [12,13]. Soriano-Disla et al. [14] hypothesized that the contents of clay and sand are more accurately estimated by MIR spectroscopy than the contents of silt because there are fundamental vibrations that are associated with clay (kaolinite (3690–3620 cm$^{-1}$), smectite (3630–3620 cm$^{-1}$), illite (3400–3300 cm$^{-1}$), and sand (quartz (1100–1000 cm$^{-1}$)); on the contrary, silt contains many minerals with more functional groups, and thus has more complex vibrations or peaks, which make them difficult to predict. The spectral bands that correspond to different soil constituents in the MIR region have been reviewed and they are presented in Table S1. In the enhancement of the soil fertility and productivity, plant nutrients, such as the available N, phosphorus (P), and potassium (K), play important

roles. Knowledge of the plant-available nutrient concentration in a piece of arable land will help to optimize the fertilizer application. Excessive fertilization affects the soil fertility and the economic investment and leads to surface and groundwater contamination [15]. It is thus crucial to improve the efficiency and accuracy of soil-available nutrient detection for optimal fertilization. However, the available nutrients in soil generally do not show any direct spectral features and usually exist in low concentrations [16]. Consequently, their quantification through the use of spectral approach through regression modeling and the validation of the models is difficult to achieve. In addition, the presence of randomly distributed irrelevant information in the spectra also greatly affects the accuracy of the calibration models for quantifying the available soil P and K contents [17].

Sophisticated statistical techniques are required for the quantitative spectral analysis of soils that use reflectance spectroscopy to uncover the response of the soil attributes from their spectra. Researchers have used various chemometric methods to relate soil spectra to soil attributes [6]. Chemometric methods are applied to extract information mathematically from the preprocessed spectral soil data. The information that is extracted is then empirically related to the conventional laboratory measurements to build the MIR calibration models [18]. MIR spectra, however, need calibration and independent validation with the laboratory-analyzed data for the development of robust prediction models for different soil properties that will be valid for a soil type [19].

In India, Alfisols and related soils cover an area of 79.7 Mha, which is about 24 percent of the country's geographical area. Alfisols are predominant in the Andhra Pradesh, Karnataka, Odisha, Chhattisgarh, Tamil Nadu, Jharkhand, Madhya Pradesh, Assam, and Uttar Pradesh states of India. These soils are dominant in the semiarid to subhumid regions of peninsular India [20]. The soils have low base saturations, low cation exchange capacities (CECs), and low organic matter contents. Kaolinite is the dominant clay mineral, and the soils are acidic in reaction. In terms of the land use and management, Alfisols are India's second most dominant soil order after Inceptisols [21]. While comparing the MIR spectral signatures of the different soil orders that are present in the United States, Zhang et al. [22] observed strong MIR absorption peaks at 3695 or 3620 cm$^{-1}$ that were due to the presence of relatively large amounts of kaolinitic minerals in the Alfisols and Ultisols. McCarty et al. [8] report excellent MIR-based predictions of soil carbonates (RMSEP = 10 g kg$^{-1}$; bias 2.5 g kg$^{-1}$) for Alfisols and Mollisols from the central United States. Although some studies on the prediction of the properties of Alfisols by MIR-diffused reflectance spectroscopy are reported [10,23], no study that explores the potential of MIR spectroscopy in conjunction with chemometrics has been conducted to predict the physical and chemical properties of Alfisols from the Indian subcontinent. The present study was undertaken to assess the potential of MIR reflectance spectroscopy in assessing the organic carbon content, the EC, the pH, the available P and K, and the sand, silt, and clay percentages in soil samples from the Alfisols regions of eastern India. The specific objectives of this study were to: (i) Evaluate the predictive performance of different algorithms for the rapid characterization of selected key soil properties by using MIR spectroscopy; (ii) Validate the models by using independent sample sets of similar soils to evaluate the predictive ability of the MIR spectroscopy in the estimation of key soil properties; and (iii) Identify spectral regions with high explanatory value for the prediction of these soil propertiesas a function of the information provided in the calibration.

## 2. Materials and Methods

### 2.1. Soil Sampling

A total of 336 random georeferenced soil samples were collected from the Alfisols of the Odisha and Jharkhand states of eastern India during April and May 2019, soon after the harvest of the winter season crops. The distribution of the sampling sites is presented in Figure 1. About 80 percent of the soil samples were collected from arable land, while the remaining 20 percent were collected from forest land. Samples were collected from the top 15 cm soil layer. Four soil subsamples, from a radius of about 20 m at each sampling

point (keeping one at the center), were collected and then mixed to obtain one composite sample for each site. Samples were carefully collected after removing surface litters and plant materials.

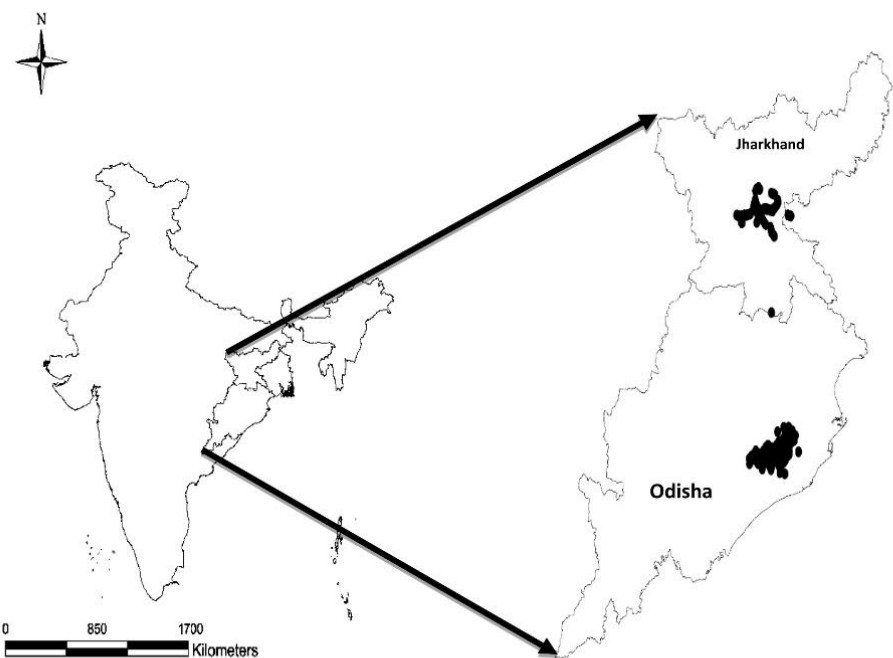

**Figure 1.** Study areas and locations of the sampling points in the states of Jharkhand and Odisha in eastern India.

### 2.2. Laboratory Analysis

The soils were air-dried at room temperature for four days to ensure stable moisture content, and large soil clods were broken and crushed gently by using a wooden rolling pin. The samples were then passed through a 2 mm sieve. The coning and quartering method was used to ensure that a homogenous subsample was selected for the wet chemistry analysis. The soil properties evaluated in this study included: the soil organic carbon (SOC); the electrical conductivity (EC); the pH; the percentages of clay, silt, and sand; and the available P and K. The EC and pH of the soil samples were determined in a 1:2.5 soil: water suspension by using the Elico Digital EC meter (Elico Pvt. Ltd., Hyderabad, India) and the Orion pH meter (Model 420A, Orion Research Inc., Franklin, MA, USA). The soil organic carbon content was determined by using the Walkley and Black wet digestion method [24]. This involves the wet combustion of the organic matter present in the soil with a mixture of potassium dichromate and sulfuric acid at about 125 °C. The soil-available P and K were analyzed following the standard procedures, as outlined in [25]. The sand, silt, and clay fractions were determined by following the hydrometer method after the pretreatment of the soil with hydrogen peroxide to remove organic matter [26].

### 2.3. MIR Spectral Measurements and Preprocessing of Soil Spectra

For the spectra measurements, soils were finely ground to an approximately <0.5 mm particle size byusing a Retsch Mortar Grinder RM 200 (Retsch, Düsseldorf, Germany). The finely ground soil samples were then loaded into sample cups, and the surfaces of the samples in the cups were leveled by using a spatula. A Bruker Alpha Fourier-Transform MIR spectrometer with a diffuse-reflectancemodule (Bruker Optics, Karlsruhe, Germany) was used to record the DRIFT spectra of the soils in the wavelength range of 4000–500 cm$^{-1}$, with a resolution of 4 cm$^{-1}$ and a zero filling of 2. This resulted in 1713 data points in each spectrum (Figure S1). During scanning of the MIR spectra, the mean of 32 internal spectral measurements was used as a representative spectrum for each soil sample. The average spectrum obtained for each sample was later transferred to a workstation, where it was first

stored in the OPUS file format provided with the instrument and was later converted to a CSV flat data table for processing and analysis. A standard gold cap was used to correct the background at the beginning of the sample scan. The background scan was then performed at an interval of about 30 min during the scanning process to correct the atmospheric and instrumental noise, and to increase the signal-to-noise ratio of the spectrum produced.

Savitzky–Golay first-derivative transformation, with a second-degree polynomial with a window size of 5 data points, was used to preprocess the absorption spectra to reduce the noise and to improve the signal-to-noise ratio. A preliminary analysis with different preprocessing treatments of the spectra indicated that the model results did not improve consistently through other preprocessing methods, such as second-derivative transformation, multiplicative scatter correction (MSC), standard normal variate (SNV) transformation, SNV detrending, and logarithm transformations. Shepherd and Walsh [19] and Towett et al. [27] also report similar findings for most of the soil variables tested across an extensive range of tropical soils. In the case of the EC and available P and K, the models developed using the square-root-transformed values performed better than the raw data. About 80 percent of the MIR spectra (*n* = 267) from the whole sample set were selected to develop the prediction models, while the remaining 20 percent (*n* = 69) of the spectra were used as the independent dataset forth validation of the models. The calibration and validation sample selection was performed following a procedure that was adapted from [28] (K–S). In the K–S algorithm, a set of samples with a uniform distribution over the predictor space is selected on the basis of a spectral distance measure. This selection encompasses all of the sources of variation found in the spectral library [29]. Here, each sample is assessed against the scores of all the samples. Though the choice of this technique does not ensure a truly independent validation set, it is nevertheless helpful in selecting one that is a useful representative [27]. For the mathematical treatments of the spectral data, we used a function written in the standard software R version 4.0 (R Core team 2020), which included the packages: hyperSpec, tidyverse, prospectr, MASS, caret, Applied Predictive Modeling, lars, pls, ggplot2, clhs, hrbrthemes, chillR, and mlbench. Before fitting for model development, the $CO_2$ absorption bands, which appeared prominently around the 2400-to-2300 wavenumber ($cm^{-1}$) regions of the MIR spectra, were excluded to eliminate the interference of $CO_2$ in the processing of the spectra.

*2.4. Chemometric Analyses*

The soil properties (raw or square-root-transformed values) and corresponding preprocessed soil spectral data were used to develop calibration models for the EC, pH, SOC content, available P, available K, and sand, silt, and clay percentages. Individual samples far from the zero line of the residual variance were deemed outliers and were excluded from the dataset. The maximum numbers of outliers (6) excluded were for the silt percentage. Three chemometric regression techniques (viz., partial least squares regression (PLSR), random forest (RF), and support vector machines (SVM)) were compared to identify the model best suited for the prediction of the soil properties. The leave-one-out cross-validation was used to calibrate the spectral data with the wet chemistry/laboratory soil data to obtain predictive models for each soil property. This is one of the most frequently used techniques for chemometric modeling and it can be considered the de facto standard method in soil spectroscopy [17]. In the PLSR technique, a large number of original descriptors are linear transformed to a new variable space on the basis of a small number of latent variables. The latent variables are chosen in such a way as to provide the minimum prediction error sum of the squares and the maximum correlation with the dependent variables [30]. The RF regression was chosen as one of the calibration methods because of its excellent ability to pick the nonlinearity relationship between the predictors and the response variables. It has also been reported to be simple in theory, fast when handling a large number of predictors, and it has an in-built fine-tuning mechanism to control overfitting. It also contains an automatic compensation mechanism for the biased sample numbers of groups during the training process [31]. The RF is reported to be a useful tool for regression studies, and it

can model both linear and nonlinear multivariate calibration [5]. On the other hand, SVM is a supervised nonparametric machine learning regression method that is effective in high-dimensional spaces and that uses a subset of training points in the decision function [32]. SVM models are sometimes more effective owing to their ability to deal with noisy patterns and the multimodal class distributions of the soil properties [33]. The tune length was set for 15 for all three approaches. The chemometric models that were developed were tested by predicting the values of a given soil variable on a validation dataset comprised of a 20 percent holdout validation soil sample. The model's performance was assessed by calculating the coefficient of determination between the predicted and observed values in the validation set ($R^2$) (Equation (1)),the root mean square error of validation (RMSE) (Equation (2)), and the ratio of the performance to the deviation (RPD) (Equation (3)), which is the ratio of the standard deviation of the validation dataset (SDval) and the standard error of prediction, which is calculated for validations that are performed with independent datasets. The coefficient of determination ($R^2$) evaluates the proportion of the total variation accounted for by the model, while the remaining variations are attributed to random error [34]. The model with higher values of $R^2$ and lower values of RMSE was selected as the best-fitted model for our study. A lower RMSE and higher $R^2$ during the model validation indicate a more accurate and robust model.

$$R^2 = \frac{\sum_{i=1}^{n}\left(\hat{y} - \bar{y}\right)^2}{\sum_{i=1}^{n}\left(y_i - \bar{y}\right)^2} \tag{1}$$

$$\text{RMSE} = \sqrt{\frac{1}{n}\sum_{i=1}^{n}\left(\hat{y} - \bar{y}\right)^2} \tag{2}$$

$$\text{RPD} = \frac{\text{SD}_{\text{val}}}{\text{RMSE}\sqrt{\frac{n}{(n-1)}}} \tag{3}$$

where $y_i$, $\bar{y}$, and represent $\hat{y}$ represent the measured values, the mean of the measured values, and the predicted values, respectively, and $n$ is the number of measurements with $i = 1, 2, \ldots, n$.

Some of the earlier studies have suggested that models with $R^2 \geq 0.75$ and an RPD $\geq 2$ provide acceptable or good accuracy levels for the prediction of the soil properties. On the other hand, models with a $R^2$ from 0.65 to 0.75, and an RPD from 1.4 to 2.0, provide satisfactory, or medium, accuracy levels of predictions [19,27,35]. The RPD values below these are considered poor for prediction. We have followed the same classification in the present study. The identification of the important wavenumbers for each soil property was performed by analyzing the distribution of the loading coefficients against the wavenumbers of the partial least-squares regression (PLSR). Higher positive or negative values of the loading coefficients at specific wavenumbers indicate the importance of these numbers in predicting the soil properties [32]. The statistical analyses and calculations were carried out using R software, version 4.0 [36].

## 3. Results and Discussion

### 3.1. Soil and Spectral Characteristics

The scattergram depicting the range, frequency, mean, and median of the soil properties is presented in Figure 2. The mean value and the data range for the soil properties were within the typical ranges for the soils in these regions [20] and covered a wide range of variability in order to promote effective model development [37]. The range for the validation dataset fell within the range of the calibration data for all the soil properties. Considerable variations in the absorbance across the spectral range were found among the samples. To demonstrate the variability in the spectra, five selected raw DRIFT spectra, which cover the spectral variability of the samples and the range of the spectral values in

each wavenumber, are depicted in Figure 3. The spectra show different high and low peaks across the spectral zones. The high peak around 1200–900 cm$^{-1}$ signifies the stretching of the Si–O–Si and Al–O–Si bonds in clay minerals, while the peaks between 850 and 580 cm$^{-1}$ denote the characteristics of the different clay minerals present [38–40]. Similarly, the strong absorbance recorded in the 3600 to 3700 cm$^{-1}$ regions of the spectra was primarily due to hydroxyl stretching vibrations that are associated with clay minerals [41]. Peaks between 3600 and 3300 cm$^{-1}$, and around 1600 cm$^{-1}$, respectively, were strongly influenced by the stretching and bending of the O–H bonds [42]. Madejova [43] also report that the spectral region above 3300 cm$^{-1}$ could be described as the OH$^-$ stretching region, while, below 1200 cm$^{-1}$ could be described as the OH- bending and Si–O stretching and bending regions. Merry and Janik [44] observed that the spectra were obtained from the mid-infrared wavelength band sense-specific molecular vibrations, which are strongly associated with the functional groups that are frequently found in soil minerals and organic matter.

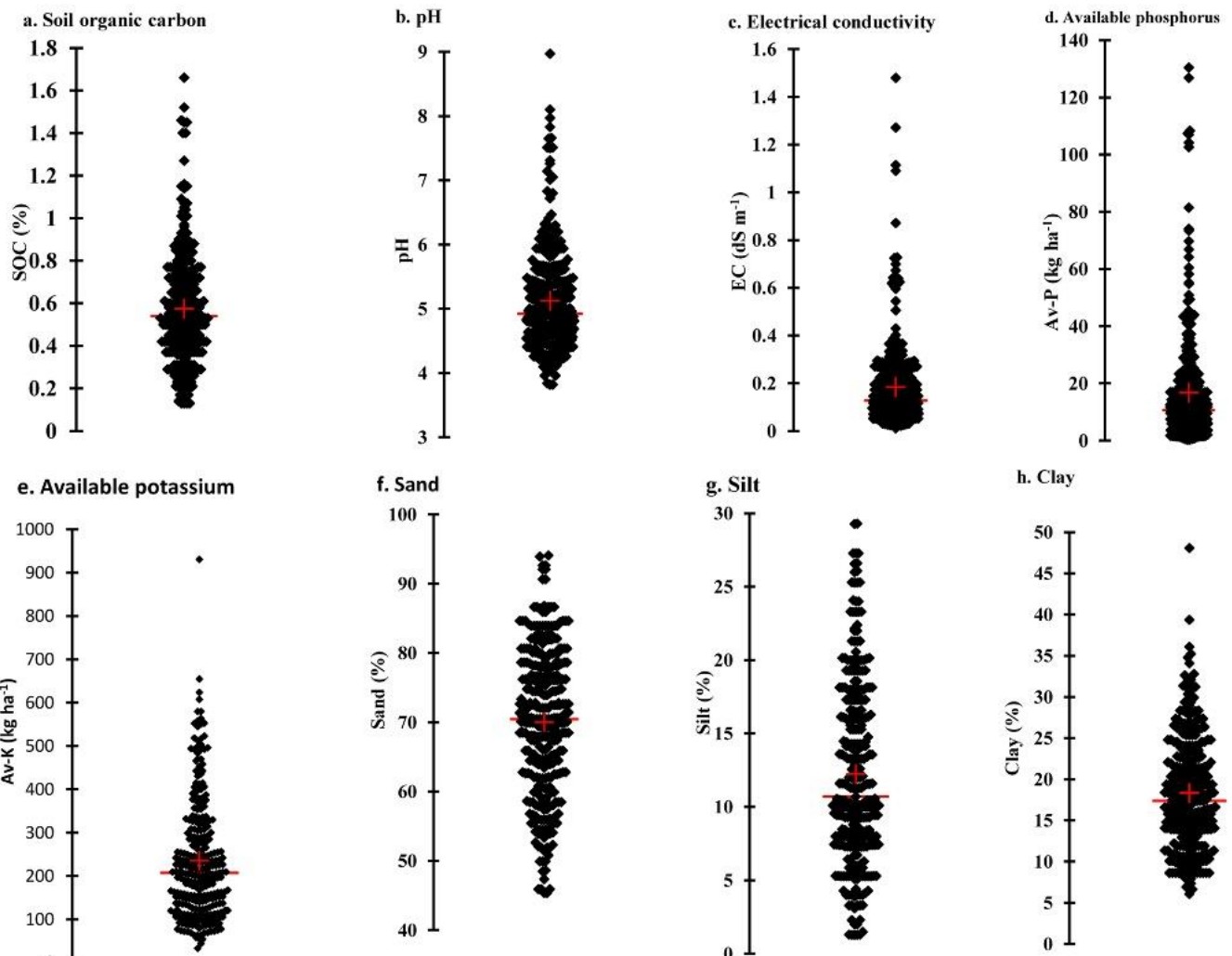

**Figure 2.** Scattergram depicting ranges, frequencies, means, and medians of: (**a**) soil organic carbon; (**b**) pH; (**c**) electrical conductivity; (**d**) available phosphorus; and (**e**) available potassium; and (**f**) sand, (**g**) silt, and (**h**) clay contents of the soil samples analyzed by standard laboratory technique.

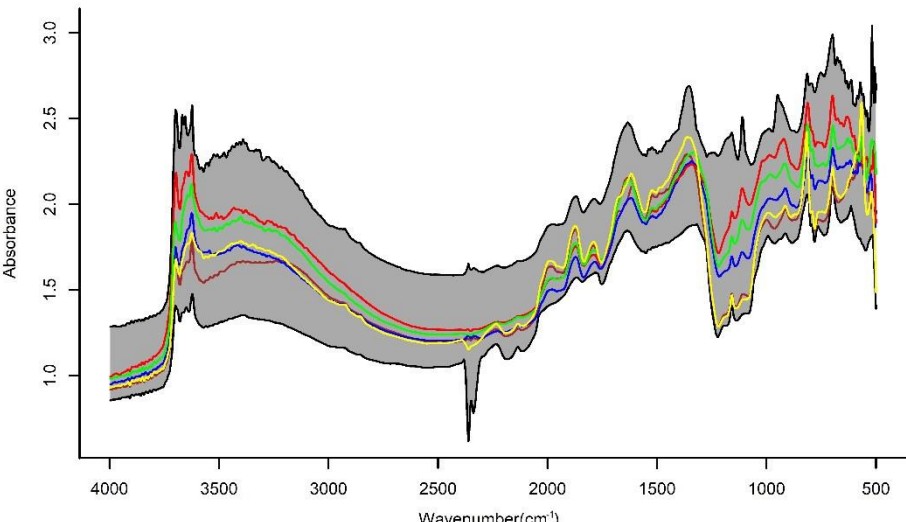

**Figure 3.** Five representative raw mid-infrared absorption spectra of the samples. Grey spectral envelope encompasses the maximum and minimum MIR absorbance values of all the soil samples at each wave number.

This indicated that the MIR spectra contained some definable peaks, which could be used in spectral interpretation to differentiate between two or more samples. Relatively subtle and broad peaks appeared near 1720, 1600 and 1370 cm$^{-1}$; this could be due to the absorptions by different types of clay minerals, the SOM, or the adsorbed moisture present in the soil [45]. The smaller peaks around 2900 cm$^{-1}$ in some of the samples were due to the aliphatic C–H stretching. Similarly, various peaks in the 3500–3000 cm$^{-1}$ region were ascribed to O–H, N–H, and C–H stretching [46]. On the other hand, the stretching and bending of various C–O, COO, and CH$_x$ bonds influenced the spectra's 1650–950 cm$^{-1}$ area [47]. However, many of the absorption peaks that arose due to the various bonds present in the SOM also overlap with the soil's mineral peaks [48].

### 3.2. Validation of SVM, PLSR, and RF Regressions with Independent Dataset

The test-set validation scatterplots of the MIR-predicted (with SVM, PLSR, and RF) versus the actual laboratory-measured values with a 1:1 line are shown in Figures 4 and 5. The scatterplots for the SOC, sand, silt, and clay were closer to 1:1 than the corresponding plots for the pH, EC, and the available P and K (Figures 4 and 5), which reveals that the ability of the MIR-spectra-based models to predict the studied soil properties varied considerably. The predictive models that were developed with PLSR techniques performed better than the two other regression techniques for all the soil properties, except for the EC, where the SVM regression model, which was developed with square-root-transformed EC values, performed best (Table 1).

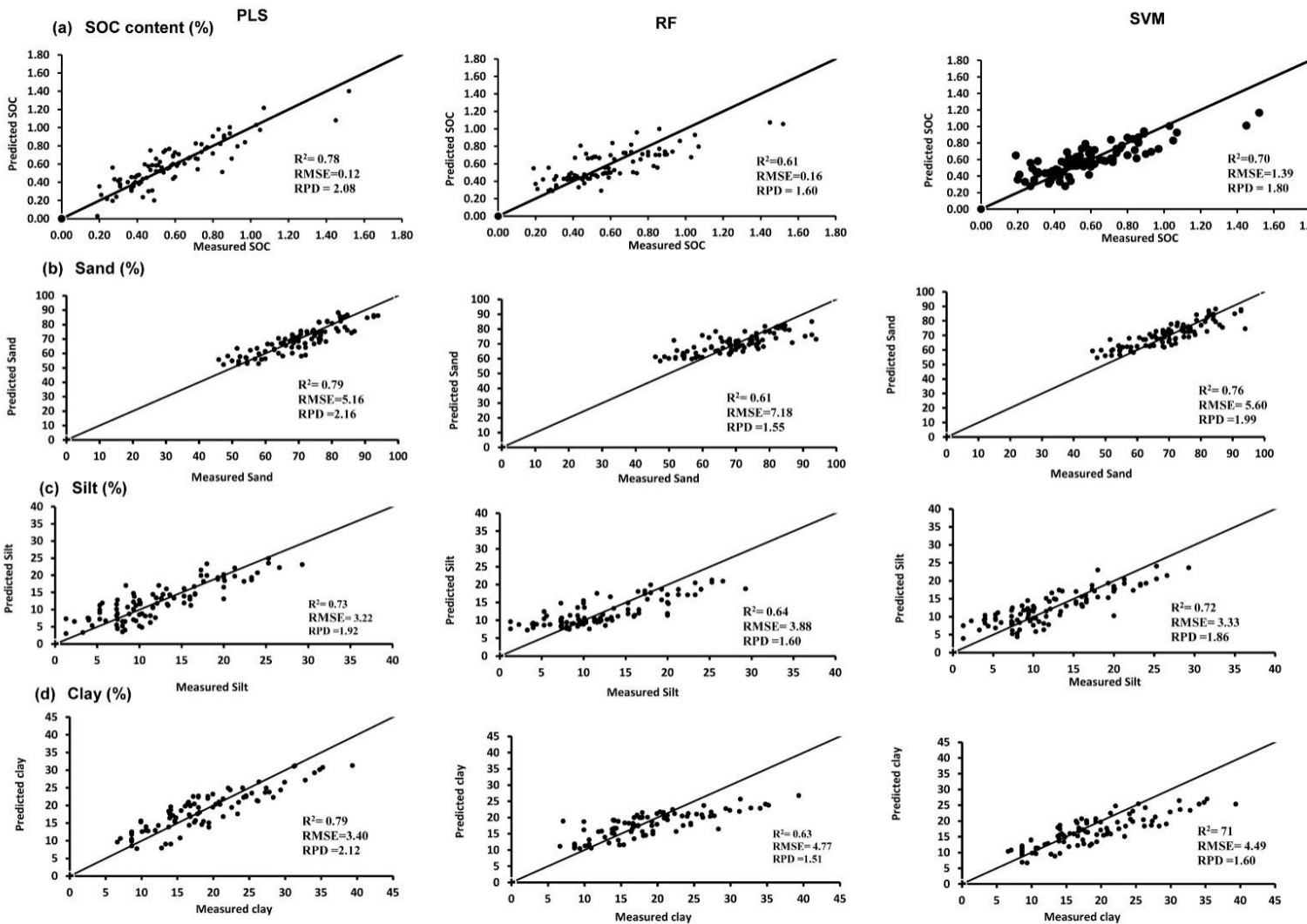

**Figure 4.** Scatterplots (1:1) of laboratory-measured versus mid-infrared diffused-reflectance-spectra-predicted values of soil properties for the random validation sample (*n* = 69) using partial least squares, random forest, and support vector machine multivariate regression models for: (**a**) soil organic carbon (SOC) content; (**b**) sand; (**c**) silt; and (**d**) clay.

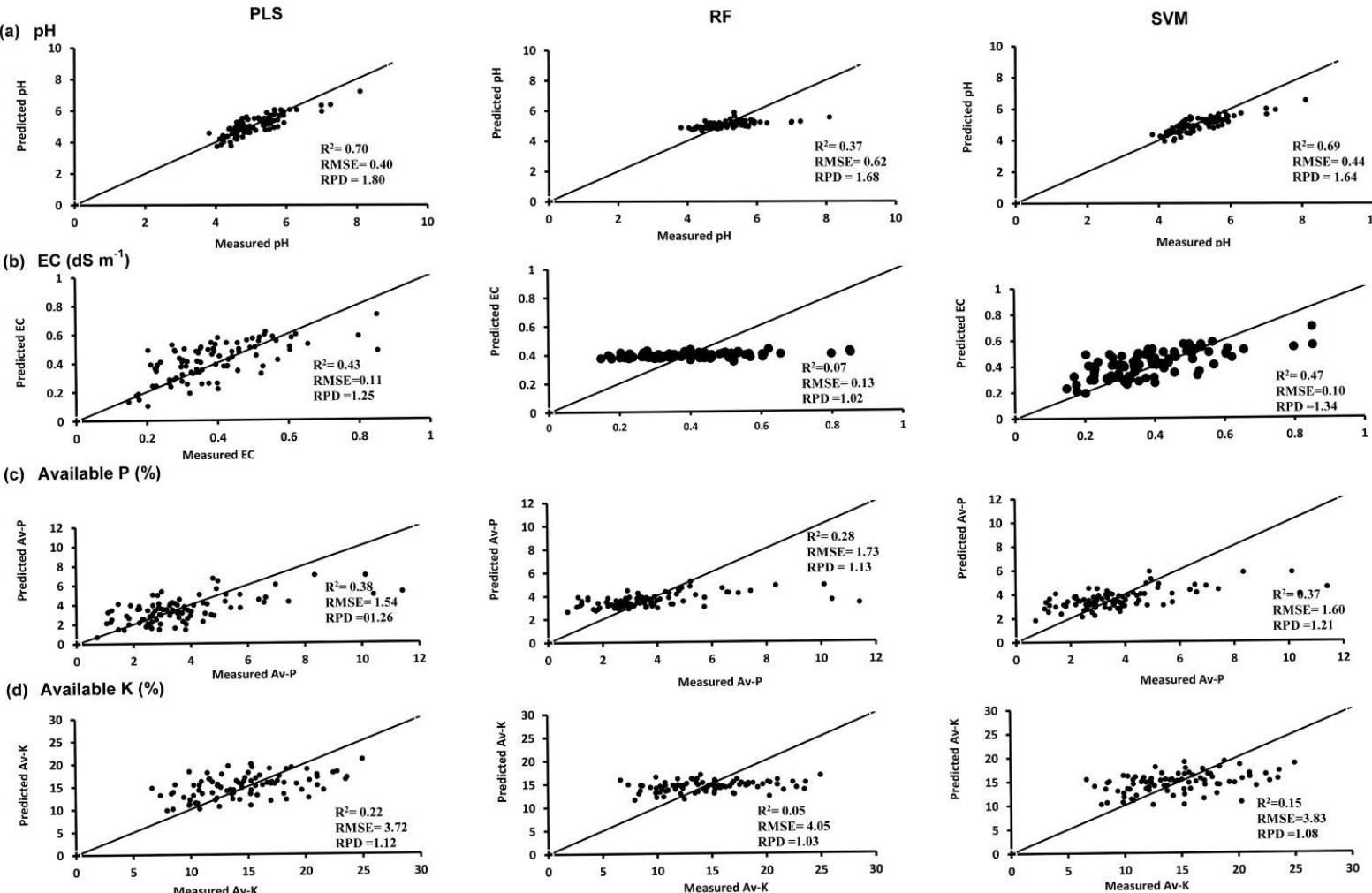

**Figure 5.** Scatterplots (1:1) of laboratory-measured versus mid-infrared diffused-reflectance-spectra-predicted values of soil properties for the random validation sample (*n* = 69) using partial least squares, random forest, and support vector machine multivariate regression models for: (**a**) pH; (**b**) electrical conductivity; (**c**) available phosphorus content; and (**d**) available potassium content.

**Table 1.** Statistics for the calibration (*n* = 267) and validation (*n* = 69) samples using partial least squares (PLS), support vector machine (SVM), and random forest (RF) regression models developed from diffused-reflectance mid-infrared spectra using the R software.

| Soil Property | Model | Calibration Set (80% of Dataset) | | | Validation Set (20% of Dataset) | | |
|---|---|---|---|---|---|---|---|
| | | $R^2$ | RMSE | RPD | $R^2_v$ | RMSEv | RPDv |
| SOC (%) | PLS | 0.88 | 0.09 | 2.85 | 0.78 | 0.12 | 2.08 |
| | SVM | 0.99 | 0.02 | 10.75 | 0.70 | 1.39 | 1.80 |
| | RF | 0.95 | 0.07 | 3.71 | 0.61 | 0.16 | 1.60 |
| pH | PLS | 0.80 | 0.36 | 2.23 | 0.70 | 0.40 | 1.82 |
| | SVM | 0.98 | 0.30 | 2.71 | 0.69 | 0.44 | 1.64 |
| | RF | 0.97 | 0.30 | 2.71 | 0.37 | 0.62 | 1.68 |
| EC * (dS m$^{-1}$) | PLS | 0.70 | 0.09 | 1.81 | 0.43 | 0.11 | 1.25 |
| | SVM | 0.94 | 0.05 | 3.42 | 0.47 | 0.10 | 1.34 |
| | RF | 0.98 | 0.07 | 2.20 | 0.07 | 0.13 | 1.02 |
| Sand (%) | PLS | 0.85 | 4.0 | 2.55 | 0.79 | 5.16 | 2.16 |
| | SVM | 0.99 | 0.97 | 10.47 | 0.76 | 5.60 | 1.99 |
| | RF | 0.96 | 2.73 | 3.73 | 0.61 | 7.18 | 1.55 |
| Silt (%) | PLS | 0.82 | 2.57 | 2.36 | 0.73 | 3.22 | 1.92 |
| | SVM | 0.99 | 0.58 | 10.34 | 0.72 | 3.33 | 1.86 |
| | RF | 0.96 | 1.63 | 3.72 | 0.64 | 3.88 | 1.60 |
| Clay (%) | PLS | 0.87 | 2.32 | 2.82 | 0.79 | 3.40 | 2.12 |
| | SVM | 0.99 | 0.63 | 10.35 | 0.71 | 4.49 | 1.60 |
| | RF | 0.95 | 1.82 | 3.58 | 0.63 | 4.77 | 1.31 |
| Available P * (kg ha$^{-1}$) | PLS | 0.70 | 1.00 | 1.83 | 0.38 | 1.54 | 1.26 |
| | SVM | 0.85 | 0.83 | 2.20 | 0.37 | 1.60 | 1.21 |
| | RF | 0.97 | 0.68 | 2.70 | 0.28 | 1.73 | 1.13 |
| Available K * (kg ha$^{-1}$) | PLS | 0.69 | 2.38 | 1.81 | 0.22 | 3.72 | 1.12 |
| | SVM | 0.98 | 0.61 | 7.07 | 0.15 | 3.83 | 1.08 |
| | RF | 0.98 | 1.62 | 2.66 | 0.05 | 4.05 | 1.03 |

* Model was developed on square-root-transformed data as the initial data were positively skewed. RMSE: root mean squared error; RPD: residual prediction deviation; $R^2$: coefficient of determination; SOC: soil organic carbon; EC: electrical conductivity.

The best models were selected on the basis of the corresponding high $R^2$ and RPD values and low RMSE values. The SVM validation models performed better than the RF models for all the soil properties studied. With decreasing sample sizes, the prediction accuracies obtained by the RF decreased more noticeably than the SVM regression for all the soil properties tested. Previous studies (e.g., [32,49]) that compare different multivariate regression models to predict the soil properties from MIR spectra have shown variable responses for different datasets and properties. While comparing the performance of different multivariate regression models in soils from the Ribeirão Inhaúma basin, Brazil, the authors of [49] obtained greater prediction accuracies with PLSR for the TOC and Mehlich−1 extractable phosphorus, whereas the SVM performed better than PLSR for clay. Deiss et al. [32] report that SVM models performed better than the PLSR for all tested properties in Tanzania and the US Midwest soils. In the present study, the RPD values for the model validation were also used to evaluate the performance of the models (Table 1). Models with an RPD > 2.0 in the validation process were considered reliable [35]. On the other hand, RPD values between 1.4 and 2 represent the model's satisfactory performance, which can be useful for the rapid screening of samples. RPD values below 1.4 were considered unacceptable [35]. The prediction models identified for the SOC (RPD = 2.06), the clay (RPD = 2.12), and the sand (RPD = 2.16) were considered reliable, while the performance of the models for the pH (RPD = 1.82) and the silt (RPD = 1.92) were satisfactory. Minasny et al. [50] report that, for agricultural applications, RPD values greater than 2 indicated that the models offered precise predictions.

The validation datasets show that the percentages of clay ($R^2$: 0.79; RMSE: 3.40) and sand ($R^2$: 0.79; RMSE: 5.16) in this study were predicted with a high degree of accuracy, followed by the SOC content ($R^2$: 0.78; RMSE: 0.12), for the models that were developed

by using the PLSR technique (Table 1). Chemical bonds of carbon-containing compounds present in the SOM contribute directly to the SOC content of the soil. The high coefficients of determination of the predictive models for the SOC are attributed to the specific strong absorption bands that are associated with these chemical bonds [51]. Urselmans et al. [52] report a similar accuracy ($R^2$ = 0.77; RPD = 2.0) for the SOC content prediction from a globally distributed soil MIR spectral library dataset. Kamau-Rewe et al. [23] also report that the SOC prediction by PLSR using DRIFT–MIR was satisfactory for the Alfisols ($R^2$ = 0.93; RMSE of cross-validation = 0.20). Recently, Olatunde [13] developed a PLSR-based chemometric prediction model for the SOC with a high accuracy ($R^2 > 0.80$), which can offer a reliable alternative to the traditional laboratory analyses. The predictions for the particle size were good for both the clay and sand percentages. The estimation of the clay and sand percentages is due to the fundamental vibrations of the associated minerals in the MIR regions [14]. Thomas et al. [53] found a good correlation between the sieve-pipette laboratory measurements and MIR-based predictions for clay ($R^2$ = 0.88) and sand ($R^2$ = 0.90) for soils from a Kenyan farm validation set. Mohanty et al. [54] also report that most of the absorption peaks that are directly or indirectly related to $SiO_2$ fall in the MIR region. Thus, sand or $SiO_2$ was predicted with greater accuracy by the MIR spectra. Urselmans et al. [52] report that the prediction of the clay content from MIR spectra was more direct, as the absorption of the spectra was primarily concentrated in the mineral regions of the spectrum. On the other hand, the important wavebands for the calibration of the sand content were distributed across the entire MIR spectrum, which suggests that the prediction for sand was relatively indirect [52]. The fingerprint region of the MIR spectrum categorically responds to quartz, and thus, the exclusion of specular reflection from the spectrum could further improve the performance of sand predictions.

The silt content ($R^2$: 0.73; RMSE: 3.22) and the pH ($R^2$: 0.70; RMSE: 0.40) were moderately predicted from the MIR spectra. Earlier studies also state that MIR spectroscopic techniques could be used to estimate the soil pH with air-dried samples [6,55]. Shepherd and Walsh [19] report a good prediction for the soil pH ($R^2$ = 0.83; RMSEC = 0.34) in their study on the characterization of the soil properties from a spectral library with 758 soils from eastern and southern Africa. Useful wavenumbers that contribute to the prediction ofthesoil pH have been associated with the functional groups, O–H, in clay minerals, water, phenols, and carboxyl and hydroxyl groups, and with $COO^-$, $CO_3^{-2}$ in carboxylates, carbonates, and carboxylic acids, which were related to the $H^+$ concentration [2]. Urselmans et al. [52] also report that important wavebands for pH predictions were mainly concentrated in the MIR spectrum parts that contain mineral features (i.e., the fingerprint and *X*–H stretching regions). However, Ji et al. [56], in their study conducted on two agricultural field soils from Canada, did not find a good prediction for the soil pH from their in situ-recorded MIR spectra.

The prediction was poor for the EC ($R^2$: 0.47; RPD: 1.25), the available P ($R^2$: 0.38; RPD: 1.26), and the available K ($R^2$: 0.22; RPD: 1.12) contents of the soil. For the available P and K contents, the best predictions were obtained when square-root-transformed values were used in the PLSR. On the other hand, the SVM regression with square-root-transformed values yielded the best prediction for the EC (Table 1). The low predictability obtained for the EC was probably due to the narrow range of EC values among the soil samples tested. Calibrations, in general, work best when there is a range of low-to-high levels for the EC [56]. In contrast, the low predictability of the extractable P and K from the MIR spectra could partially be due to the fact that the extracted amounts of P and K do not correlate significantly with the soil properties that are MIR-active [27]. The predictability of the extractable P and K from the MIR spectra of air-dried samples was also reported to be poor in previous studies [6,55,57], owing to their poor MIR spectral signatures [56]. Wijewardane et al. [17], from their study using US soil spectral library samples, concluded that properties such as K and P cannot be assigned to a particular MIR absorption band, which makes these properties challenging to predict, unlike other properties. Similarly, Minasny et al. [50] did not find acceptable values in predicting the available P from the

MIR spectrum. Recently, Ma et al. [2], while analyzing the performance of three MIR spectroscopic techniques (viz., diffuse reflectance spectroscopy (DRF), attenuated total reflectance spectroscopy (ATR), and photoacoustic spectroscopy (PAS)) found that all three spectral techniques poorly predicted the available phosphorus contents because of a weak correlation between the absorption and the molecular vibration of the phosphorus in the mid-infrared region. The predictability of the available K in soils is poor because the MIR spectral region does not have adequate information to accurately predict the K content in soils, and particularly the extractable K [14]. Besides this, the extractable K concentration in the soil solution is relatively low compared to the Ca and Mg concentrations and, consequently, the effect of the $K^+$ on the soil spectra might have been largely concealed or shrouded by those for the $Ca^{2+}$ and $Mg^{2+}$ [14,50]. On the other hand, Bonett et al. [7] reason that the poor predictability of the available K was primarily due to the high mobility of the positively charged K ions in the soil solution, which easily varied their contents, which caused less certain prediction results. Dimkpa et al. [58] also report that spectroscopy-based testing methods are not suitable for the estimation of the soil nutrient fractions that are potentially bioavailable to the crops. Here, the chemicals that were used for the extraction of the elements from the soil give some indication of the bioavailable elemental concentration.

### 3.3. Identifying Wavenumbers with High Explanatory Value

Plots that represent the wavenumber versus the PLSR loading coefficient for the models that were developed for the SOC, the pH, and the clay and sand contents are depicted in Figure 6. Wavenumbers with higher loading coefficients indicate a greater weightage to the final predicted values [59], which offer spectroscopy-based explanations of these models. The important wavebands that were identified for the prediction of the SOC were: 1225–1223; 1462–1480; 1505–1509; 1523–1530; 1747–1758; 2085–2090; 2155–2170; 2836–2851; and 2905–2915 $cm^{-1}$ (Figure 6a).

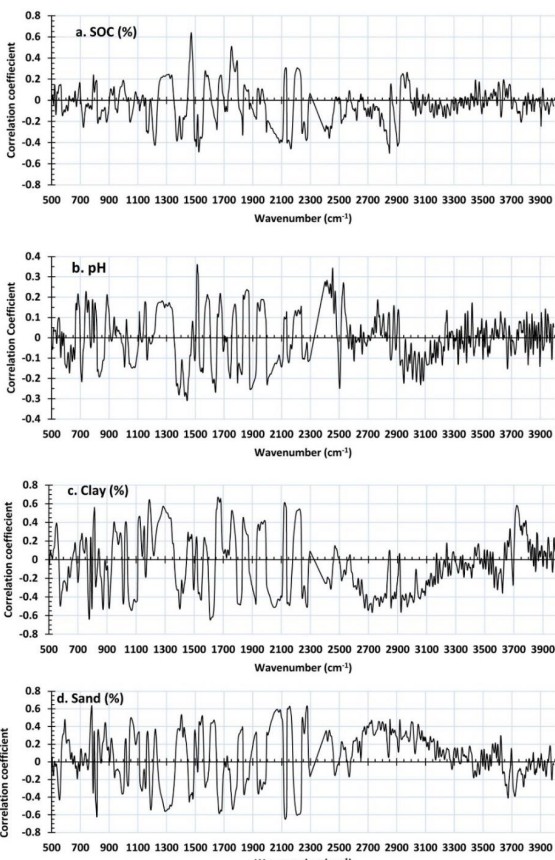

**Figure 6.** Spectral distributions of loading coefficients of the partial least-squares regression (PLSR) model for: (**a**) soil organic carbon content; (**b**) pH; (**c**) clay; and (**d**) sand.

The C–H stretching at 2910 and 2850 cm$^{-1}$, along with the C = O stretching at 1750 cm$^{-1}$ [2], appeared to be higher in the SOC model. However, Urselmans et al. [52] observed that the important MIR wavebands for the SOC estimation are present across the full spectral range, from: 746 to 663; 1047 to 1041; 1139 to 1105; 1274 to 1265; 1573 to 1560; 1820 to 1881; 2547 to 2524; 2624 to 2611; 2985 to 2923; 3370 to 3353; 3683 to 3666; and 3783 to 3733 cm$^{-1}$. Ma et al. [2] found that the effective wave numbers for the SOM values were generally associated with the carbon functional groups and also with the absorption of iron oxides. The important wavebands that were identified for the clay and sand calibration were concentrated in the mineral region of the spectrum [52]. The peaks that were obtained from the present study show that PLSR models could extract valuable information related to different soil constituents since the peaks that were obtained were highly correlated with the theoretically identified spectral zones.

## 4. Conclusions

Our study indicates that the PLSR models performed better than the RF- and SVM-based chemometric models in estimating most of the soil properties. The SOC, pH, and the sand and clay contents of the soils were well predicted by using both the PLSR and SVM models; however, they failed to predict the extractable nutrients (viz., the available P and K contents of the soil) with reasonable accuracy. We also identified specific wavebands that contributed to the prediction of the soil SOC, the pH, and the clay and sand contents. MIR spectroscopy showed great potential for the simultaneous estimation of the SOC, the pH, and the particle size distribution of the Alfisols from eastern India. However, the models that were developed in this study need to be validated beyond these locations for a more robust and stable prediction. The spectral bands that were identified in this research will help with understanding the underlying phenomenon that contributes to the improved estimation of specific soil properties. This will assist in the development of cost-effective hand-held sensors for nutrient recommendation/application. However, in-depth studies are warranted to assist in understanding the inherent associations of MIR reflectance with different soil properties, for which MIR can be used as a reliable and precise method for the measurement vis-à-vis the soil health assessment and management.

**Supplementary Materials:** The following supporting information can be downloaded at: https://www.mdpi.com/article/10.3390/su14094883/s1, Figure S1: A representative picture of MIR instrument with Drift spectra; Table S1: The characteristics band assignment of various soil constituents in mid infrared region.

**Author Contributions:** Conceptualization: K.M.H., N.K.S., J.R., A.K.P. and S.K.C.; Methodology: K.M.H., N.K.S., M.M., P.J. and P.D.; Software: N.K.S., K.M.H., A.S., E.T., S.L., K.S. and E.W.; Validation: K.M.H., N.K.S., P.D., M.V.C., S.J. and P.M.; Formal analysis: K.M.H., N.K.S., P.J. and P.M.; investigation, K.M.H., R.S.C., M.M., P.J., J.K.T.; Resources, M.S., S.K.D., C.B.; data curation, N.K.S., K.M.H., M.M., P.J. and R.S.C., writing—original draft preparation, K.M.H., N.K.S., M.M. and R.S.C.; writing—review and editing: S.J., S.L., J.R., A.K.P., S.K.C., S.K.D., C.B. and K.S.; Visualization: K.M.H., N.K.S., A.S., E.T. and E.W.; supervision, A.K.P., S.K.C., J.R. and S.K.D.; project administration, K.M.H., A.K.P., S.K.C., J.R. and S.K.D.; funding acquisition, A.K.P., S.K.C., J.R. and S.K.D. All authors have read and agreed to the published version of the manuscript.

**Funding:** This research is jointly funded by World Agroforestry (ICRAF), Nairobi, Kenya, and ICAR, India.

**Institutional Review Board Statement:** Not applicable.

**Informed Consent Statement:** Not applicable.

**Data Availability Statement:** Not applicable.

**Conflicts of Interest:** The authors declare that they have no conflict of interest.

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
