# Peer review of "Mid-Infrared Reflectance Spectroscopy for Estimation of Soil Properties of Alfisols from Eastern India"

_sustainability, doi:10.3390/su14094883_

Round 1
Reviewer 1 Report
The manuscript presents an interesting evaluation of soil properties. It should be of interest of readers devoted to this topic.
My comments are the following:
References need to be updated to more recent dates
Figures 4 and 5 are not well visible, is necessary to arrange them
Author Response
Dear Sir,
Greetings
A apoint-by-point response to the reviewer’s comments is attached here.
Thank you

Reviewer 2 Report
Great paper; just a few minor points that could be improved (see attachment).

Author Response
Dear Sir,
Greetings
A point-by-point response to the reviewer’s comments is attached here.
Thank you

Reviewer 3 Report
In the present paper, the authors report the results of their investigation on the properties of a particular soil in Easter India using MIR reflectance spectroscopy. The infrared spectra of many samples were collected and calibration procedures using different regression techniques were compared.
The topic addressed by the authors is interesting and the paper is well-organized. Before publication, I suggest adding further information that would help readers not particularly skilled in MIR spectroscopy. In particular, to describe in detail one representative spectrum of Alfisols with the indication of the main peaks and bands with their assignment, perhaps a table would be useful. Moreover, the authors should evidence the eventual improvements offered by their approach in comparison to the older ones.
A certain number of other minor changes are reported in the following list.
a) Please clarify what is the meaning of “zero filling of 2” at line 174.
b) A photo or a schematic drawing of the geometry adopted for collecting DRIFT spectra would be useful in the 2.3 section.
c) What are exactly the “square root transformed values” cited in line 192? Do the authors have any idea because they work well in some cases?
d) In the Caption of Figure 2 please specify that the pictograms report data obtained with conventional experimental techniques.
e) As I said before I suggest showing in Figure 3 a representative spectrum with the indication of peaks and to add a table with the assignment.
f) In figure 6 the authors report the spectral distribution of correlation coefficients of the partial least-squares regression model for soil, organic carbon content, pH, clay and sand. Why do they not show the correlation coefficients for the other examined properties?
g) Please carefully revise the English language because some sentences are not clear (e.g., see lines 108-110, 126-128, 179-181)
Author Response

(The authors gave the same response as above.)
